# Effect of Oscillation and Pulmonary Expansion Therapy on Pulmonary Outcomes after Cardiac Surgery [†]

Christopher D. Williams [1], Kirsten M. Holbrook [1], Aryan Shiari [1,*], Ali A. Zaied [1], Hussam Z. Al-Sharif [1], Abdul R. Rishi [2], Ryan D. Frank [3], Adel S. Zurob [1] and Muhammad A. Rishi [4]

[1] Pulmonary, Critical Care, and Sleep Medicine, Mayo Clinic Health System—Northwest Wisconsin Region, 1221 Whipple St., Eau Claire, WI 54703, USA; williams.christopher1@mayo.edu (C.D.W.); holbrook.kirsten@mayo.edu (K.M.H.); zaied.ali@mayo.edu (A.A.Z.); al-sharif.hussam@mayo.edu (H.Z.A.-S.); zurob.adel@mayo.edu (A.S.Z.)
[2] Department of Internal Medicine, SSM Health St Clare Hospital, Fenton, MO 63026, USA; rishi.abdul@mayo.edu
[3] Division of Clinical Trials and Biostatistics, Mayo Clinic, Rochester, MN 55905, USA; frank.ryan@mayo.edu
[4] Division of Pulmonary, Critical Care, Sleep and Occupational Medicine, Indiana University School of Medicine, Indianapolis, IN 46202, USA; mrishi@iu.edu
\* Correspondence: shiari.aryan@mayo.edu
[†] Mayo Clinic does not endorse specific products or services included in this article. Presented as a poster at the American Association of Respiratory Care (AARC) Congress; 9–12 November 2019; New Orleans, Louisiana. Portions of this manuscript have been published in abstract form: Respiratory Care. 2019 Oct; 64 (Suppl 10): 3216240.

**Abstract:** Background: Oscillation and pulmonary expansion (OPE) therapy can decrease postoperative pulmonary complications in a general surgical population, but its effect after cardiac surgery has not been reported, to our knowledge. We hypothesized that using an OPE device after cardiac surgery before extubation would decrease pulmonary complications. Methods: This retrospective cohort study included adults undergoing elective open cardiac surgery at our institution from January 2018 through January 2019, who had an American Society of Anesthesiologists score of 3 or greater. For mechanically ventilated patients after cardiac surgery, a new OPE protocol was adopted, comprising an initial 10-min OPE treatment administered in-line with the ventilator circuit, then continued treatments for 48 h after extubation. The primary outcome measure was the occurrence of severe postoperative respiratory complications, including the need for antibiotics, increased use of supplemental oxygen, and prolonged hospital length of stay (LOS). Demographic, clinical, and outcome data were compared between patients receiving usual care (involving post-extubation hyperinflation) and those treated under the new OPE protocol. The primary outcome measure was the occurrence of severe postoperative respiratory complications, including the need for antibiotics, increased use of supplemental oxygen, and prolonged hospital length of stay (LOS). Demographic, clinical, and outcome data were compared between patients receiving usual care (involving post-extubation hyperinflation) and those treated under the new OPE protocol. Results: Of 104 patients, 54 patients received usual care, and 50 received OPE. Usual-care recipients had more men (74% vs. 62%; $p = 0.19$) and were older (median, 70 vs. 67 years; $p = 0.009$) than OPE recipients. The OPE group had a significantly shorter hospital LOS than the usual-care group (mean, 6.2 vs. 7.4 days; $p = 0.04$). Other measures improved with OPE but did not reach significance: shorter ventilator duration (mean, 0.6 vs. 1.1 days with usual care; $p = 0.06$) and shorter LOS in the intensive care unit (mean, 2.7 vs. 3.4 days; $p = 0.06$). On multivariate analysis, intensive care unit LOS was significantly shorter for the OPE group (mean difference, $-0.85$ days; 95% CI, $-1.65$ to $-0.06$; $p = 0.04$). The OPE group had a lower percentage of postoperative complications (10% vs. 20%). Conclusions: OPE therapy after cardiac surgery is associated with decreased intensive care unit (ICU) and hospital LOS.

**Keywords:** cardiothoracic surgery; continuous high-frequency oscillation; pneumonia; postoperative pulmonary complications

## 1. Introduction

Respiratory complications after surgery have a substantial burden on patient outcomes and health care costs [1]. These complications include lower respiratory tract infection, acute respiratory failure, atelectasis and persistent pneumothorax, need for prolonged mechanical ventilation, prolonged intensive care unit (ICU) stay, and extubation failure. The surgical site affects rates of pulmonary complications, which are more common among patients who undergo cardiothoracic, thoracic, and upper abdominal surgery. The incidence of pulmonary complications varies from 2% to 5% in a general surgical population, from 3% to 16% after coronary artery bypass graft (CABG) surgery, and from 5% to 7% after valvular heart surgery [2–5]. Other risk factors for pulmonary complications include older age, a higher American Society of Anesthesiologists (ASA) Physical Status Classification score (ranging from ASA I—A normal healthy patient to ASA-VI A declared brain-dead patient whose organs are being removed for donor purposes), congestive heart failure, chronic obstructive pulmonary disease, smoking history, and severe (class 3) obesity [6–10].

Atelectasis is a major factor in developing other postoperative pulmonary complications [11]. Although most patients in a previous study had atelectasis after surgery, perioperative interventions addressing atelectasis in high-risk patients were shown to decrease the risk of pulmonary complications including respiratory failure [12]. Among the approaches shown to decrease postoperative atelectasis are adequate and judicious analgesia and nasogastric decompression for carefully selected patients [11,12]. High-risk patients may benefit from pulmonary secretion mobilization and pulmonary inflation interventions [11–14]. Devices shown to improve pulmonary inflation include those that provide continuous positive airway pressure (PAP) and those that use oscillation and pulmonary expansion (OPE) [15–21]. Whereas PAP devices improve hypoxemia in addition to atelectasis [22,23], OPE devices help clear mucus, promote lung expansion, and can be used for nebulization [15,24–29].

While a prospective study suggested that aggressive treatment with OPE may reduce postoperative pulmonary complications and resource utilization in high-risk patients undergoing general surgery, including a small subset undergoing thoracic surgery, its impact after cardiac surgery remains unexplored in the literature [30]. To address this gap, the current study aims to investigate whether OPE can effectively decrease the incidence of postoperative respiratory complications in high-risk patients undergoing cardiac surgery compared to those receiving standard care.

## 2. Methods

The Institutional Review Board (IRB) approval mentioned herein pertains to the original study conducted, and not to the manuscript of this current paper. The study was approved by the Mayo Clinic IRB for the use of existing medical records of patients who gave prior research authorization. Specifically, the original study received approval on 28 January 2019, from the Mayo Clinic IRB for the use of existing health records of patients who had provided prior research authorization. The IRB assessed and determined that this original study's activities did not necessitate review in accordance with the Code of Federal Regulations (45 CFR 46.102), hence no IRB number was assigned.

### 2.1. Study Design

Approval for the study was obtained in anticipation of the planned introduction of the new protocol in the ICU in 2019. Data was then collected retrospectively upon completion of each study phase (Usual care from March to June 2019 vs. OPE therapy from July to October 2019). Consistent with principles of TREND reporting guidelines for Quasi-Experimental Study Designs [31], we performed a retrospective health record review of all consecutive patients 18 years or older with an ASA score of 3 or greater undergoing elective CABG, mitral valve replacement (MVR), and aortic valve replacement (AVR) surgery from 1 March 2019, through 31 October 2019, at a community hospital in Northwest Wisconsin. Only open elective surgical procedures were included. Patients

were excluded from analysis if they had a contraindication to OPE therapy (e.g., untreated tension pneumothorax), underwent a minimally invasive procedure, received ventilator therapy before surgery, or had a history of home PAP use.

Demographic, clinical, and outcomes data were collected for study participants. Data collected included ICU length of stay (LOS), hospital LOS, duration of mechanical ventilation, and the rate of all complications occurring during hospitalization, including for lower respiratory tract infections.

## 2.2. Study Device

The OPE device used was the MetaNeb System (Hillrom, Chicago, IL, USA). The device has a pneumatic compressor that administers continuous high-frequency oscillation and continuous positive expiratory pressure. This system was developed for mobilizing respiratory secretions, expanding lungs, and preventing and treating atelectasis. The device can also be used for delivering nebulization while it is in continuous high-frequency oscillation or continuous positive expiratory pressure mode [32].

## 2.3. Treatment Regimen

From 1 March through 30 June 2019, consecutive patients undergoing these procedures received either incentive spirometry after extubation according to a nursing protocol or PAP (EzPAP, Smiths Medical ASD) according to a respiratory therapy protocol, or both. The choice of intervention was based on the attending physician's preference. For both protocols, patients were instructed to breathe through the PAP device mouthpiece for 10 consecutive breaths, with a target expiratory pressure of 15 cm $H_2O$. At the end of this breathing cycle, patients breathed normally for 1 min. Then this process of targeted breathing and eupnea was repeated 3 times. To help patients reach a target expiratory pressure of 15 cm $H_2O$ during lung expansion therapy, the oxygen gas flow meter was adjusted to inspiratory flows of 5 to 12 L/min.

On 1 July 2019, our department adopted a new protocol that universally incorporated OPE treatment for mechanically ventilated patients undergoing CABG, AVR, or MVR surgery who had an ASA score of 3 or greater. Patients were transferred from the operating room to the critical care unit. Within 2 h after patients were deemed hemodynamically stable while receiving mechanical ventilation, a 10-min OPE treatment was administered in-line with the ventilator circuit. After extubation, patients continued to receive incentive spirometry but no longer received PAP therapy during OPE treatment. Extubated patients were given OPE treatments 4 times daily for 48 h and then were reevaluated. If a patient had a vital capacity of 15 mL/kg or greater, the protocol was discontinued. Nebulizer treatment was not to be delivered during OPE sessions. All patients were extubated according to an extubation protocol for cardiothoracic surgery (Figure 1).

## 2.4. Outcome Measures

Our primary outcome measure was development of severe postoperative respiratory complications. Postoperative respiratory complications that patients were screened for included the need for prolonged mechanical ventilation (>24 h after postsurgical hospital admission), prolonged need for noninvasive positive pressure ventilation (>24 h after hospital admission), prolonged increased oxygen requirements (>40% fraction of inspired oxygen or 5 L/min >24 h after admission), and readmission to the ICU. Screening also included a diagnosis of pneumonia based on criteria [31] consisting of new pulmonary infiltrate, new-onset fever, purulent sputum, leukocytosis, and increased oxygen requirements. A positive result from a sputum culture was not required for the diagnosis. Other outcomes were duration of mechanical ventilation, ICU LOS, and hospital LOS.

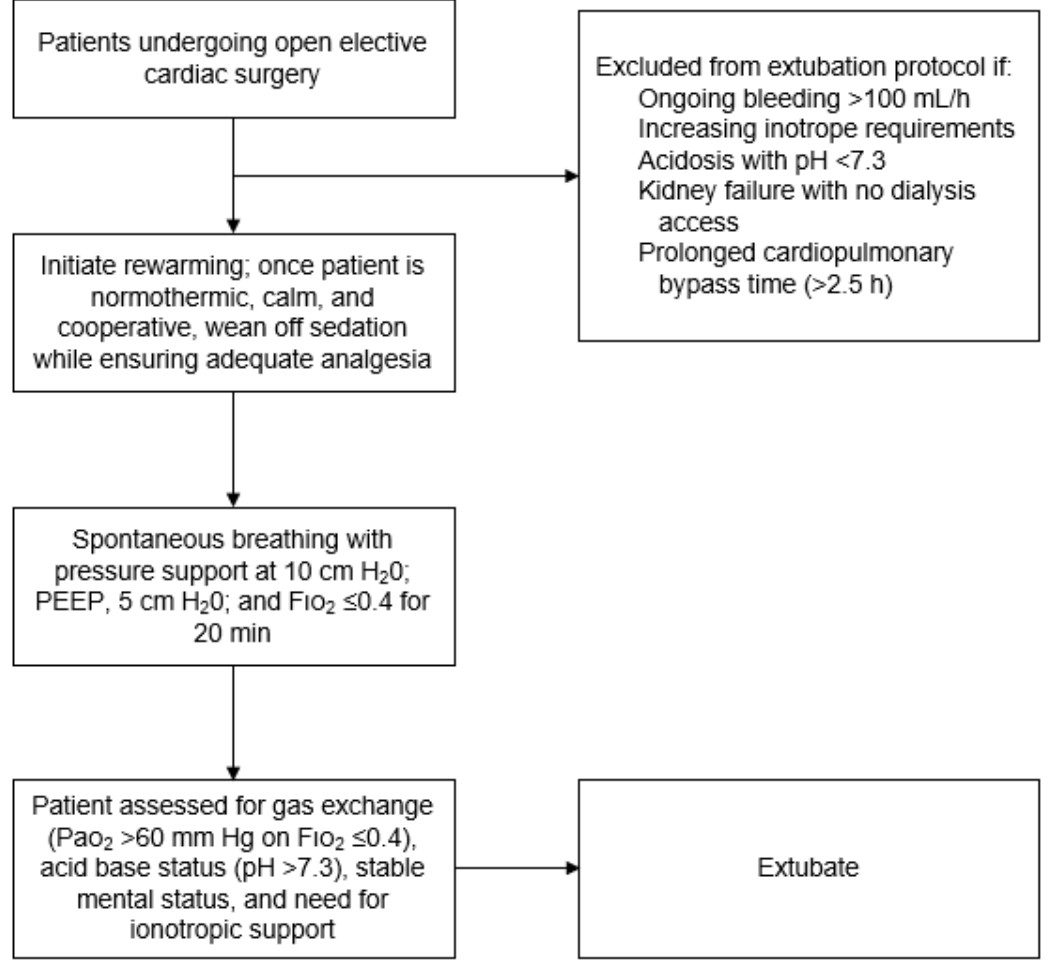

**Figure 1.** Extubation protocol after elective cardiac surgery. $F_{IO_2}$ indicates fraction of inspired oxygen; $Pa_{O_2}$, arterial partial pressure of oxygen; PEEP, positive end-expiratory pressure.

*2.5. Statistical Analysis*

Analysis was performed with SAS version 9.3 (SAS Institute Inc. *Madison 222 West Washington Ave. Suite 470. Madison, WI 53703*). All hypothesis tests were 2-tailed, with $p \leq 0.05$ considered significant. Patients' demographic characteristics and primary and secondary outcomes were summarized with descriptive statistics: number (%) for categorical variables and mean (SD) or median (IQR) for continuous variables. The Wilcoxon rank sum test was used to compare continuous variables, and the $\chi^2$ test or the Fisher exact test was used to compare categorical variables. Univariate and multivariate associations between the treatment phase and outcomes were further defined by using linear and multiple logistic regression models where appropriate to obtain mean differences or odds ratios.

**3. Results**

In total, 104 adults undergoing cardiac surgery who had an ASA score of 3 or greater were studied from March 2019 through October 2019. Of these patients, 54 received usual care before the OPE intervention, and 50 received OPE therapy after the new protocol was implemented (Figure 2).

The usual-care group was older than the OPE group (median age, 70 vs. 67 years; $p = 0.009$) and had more men (74% vs. 62%; $p = 0.19$), but no other difference between study groups was observed in demographic characteristics or in preoperative risk according to ASA score (Table 1). The distribution of surgical procedures performed before and after intervention also was similar. With OPE treatment, hospital LOS was significantly shorter than with usual care (mean, 6.2 vs. 7.4 days; $p = 0.04$; Tables 2 and 3). Although ventilator duration tended to be shorter for the OPE group, this difference did not reach significance (mean, 0.6 vs. 1.1 days; $p = 0.06$); nor did the shorter ICU LOS observed after intervention (mean, 2.7 vs. 3.4 days; $p = 0.09$). No difference was observed in duration of oxygen use before and after intervention (mean, 3.6 vs. 4.2 days; $p = 0.99$).

**Table 1.** Baseline characteristics by study phase [a].

| Characteristic | Total (N = 104) | Usual Care [b] (n = 54) | OPE Therapy [c] (n = 50) | *p*-Value [d] |
|---|---|---|---|---|
| Age, y | 70 (64–77) | 73 (66–78) | 67 (62–74) | 0.009 [e] |
| Sex | | | | 0.19 |
| Men | 71 (68) | 40 (74) | 31 (62) | |
| Women | 33 (32) | 14 (26) | 19 (38) | |
| ASA score | | | | 0.85 |
| 3 | 20 (19) | 10 (19) | 10 (20) | |
| 4 | 84 (81) | 44 (82) | 40 (80) | |
| Any CABG | 69 (66) | 36 (67) [f] | 33 (66) | 0.94 |
| Any AVR | 31 (30) | 16 (30) | 15 (30) | 0.97 |
| Any MVR | 5 (5) | 3 (6) | 2 (4) | 1.00 [g] |

Abbreviations: ASA, American Society of Anesthesiologists; AVR, aortic valve replacement; CABG, coronary artery bypass graft; MVR, mitral valve replacement; OPE, oscillation and pulmonary expansion. [a] Data are number (%) except for age, which is reported as median (IQR). [b] Usual care included incentive spirometry and positive airway pressure therapy as needed. [c] OPE was delivered by the MetaNeb System (Hillrom). [d] A $\chi^2$ test was used unless otherwise indicated, *p* value < 0.05 considered significant; [e] Wilcoxon rank sum test. [f] One patient in the usual-care group underwent >1 surgical procedure. [g] Fisher exact test.

Although the overall complication rate did not significantly differ before and after intervention (Table 4), a decrease was observed in the rate of all respiratory tract infections after intervention. Specifically, no cases of postoperative pneumonia developed in the OPE group compared with four cases in the usual-care group. No adverse events were reported related to the device.

**Table 2.** Outcomes by study phase.

| Characteristic | Total (N = 104) | Usual Care [a] (n = 54) | OPE Therapy [b] (n = 50) | *p*-Value |
|---|---|---|---|---|
| Ventilator duration, median (IQR), d | 0.5 (0.5–0.5) | 0.5 (0.5–1.0) | 0.5 (0.5–0.5) | 0.06 [c] |
| Hospital LOS, median (IQR), d | 6 (5–8) | 6 (5–8) | 6 (4–7) | 0.04 [c] |
| ICU LOS, median (IQR), d | 2 (2–4) | 3 (2–4) | 2 (2–3) | 0.09 [c] |
| Oxygen duration, median (IQR), d | 3 (2–5) | 3 (2–5) | 3 (2–4) | 0.99 [c] |
| PAP (EzPAP, Smiths Medical ASD) or hyperinflation, No. (%) | 47 (45.2) | 47 (87.0) | 0 (0) | <0.001 [d] |
| Any complication, No. (%) | 16 (15.4) | 11 (20.4) | 5 (10.0) | 0.14 [d] |
| Infection, No. (%) | 5 (4.8) | 5 (9.3) | 0 (0) | 0.03 [d] |

Abbreviations: ICU, intensive care unit; LOS, length of stay; OPE, oscillation and pulmonary expansion; PAP, positive airway pressure. [a] Usual care included incentive spirometry and PAP therapy as needed. [b] OPE was delivered by the MetaNeb System (Hillrom). [c] Wilcoxon rank sum test, *p*-value < 0.05 considered significant. [d] $\chi^2$ test, *p* value < 0.05 considered significant.

**Figure 2.** Configuration of the participants.

**Table 3.** Univariate and multivariate associations between study phase and continuous outcomes with linear regression.

| Outcome | N | Mean (SD) | Univariate Analysis [a] | | Multivariate Analysis [b] | |
|---|---|---|---|---|---|---|
| | | | Mean Difference (95% CI) | *p*-Value | Mean Difference (95% CI) | *p*-Value |
| Ventilator duration, d | | | | 0.08 | | 0.13 |
| Usual care | 54 | 1.1 (1.8) | 0.0 [Reference] | | 0.0 [Reference] | |
| OPE therapy | 50 | 0.6 (0.4) | −0.44 (−0.94 to 0.05) | | −0.40 (−0.92 to 0.11) | |
| Hospital stay, d | | | | 0.04 | | 0.10 |
| Usual care | 54 | 7.4 (3.7) | 0.0 [Reference] | | 0.0 [Reference] | |
| OPE therapy | 50 | 6.2 (2.4) | −1.27 (−2.47 to −0.06) | | −1.04 (−2.26 to 0.18) | |
| ICU stay, d | | | | 0.06 | | 0.04 |
| Usual care | 54 | 3.4 (2.5) | 0.0 [Reference] | | 0.0 [Reference] | |
| OPE therapy | 50 | 2.7 (1.3) | −0.74 (−1.52 to 0.03) | | −0.85 (−1.65 to −0.06) | |
| Oxygen duration, d | | | | 0.34 | | 0.51 |
| Usual care | 54 | 4.2 (3.9) | 0.0 [Reference] | | 0.0 [Reference] | |
| OPE therapy | 50 | 3.6 (2.1) | −0.58 (−1.78 to 0.62) | | −0.41 (−1.64 to 0.82) | |

Abbreviations: ICU, intensive care unit; OPE, oscillation and pulmonary expansion. [a] Regression models included only treatment phase, *p* value < 0.05 considered significant; [b] Regression models included treatment phase, age, sex, and American Society of Anesthesiologists score, *p*-value < 0.05 considered significant.

After multivariate adjustment for potential confounders (including study phase, age, sex, and ASA score), ICU LOS was significantly shorter after intervention (mean difference, −0.85 days; 95% CI, −1.65 to −0.06 days; *p* = 0.04; Table 3). The OPE group also had a lower percentage of complications than the usual-care group (10% vs. 20%), but the difference was not significant on multivariate analysis (odds ratio [95% CI] = 0.51 [0.15–1.66]; *p* = 0.26).

**Table 4.** Postoperative complications by study phase, No. (%).

| | Total (n = 104) | Usual Care [a] (n = 54) | OPE Therapy [b] (n = 50) | *p*-Value [c] |
|---|---|---|---|---|
| Complication | | | | |
| None | 88 (85) | 43 (80) | 45 (90) | |
| Pneumonia | 4 (4) | 4 (7) | 0 (0) | |
| NIV | 2 (2) | 1 (2) | 1 (2) | |
| MV | 1 (1) | 1 (2) | 0 (0) | |
| Tracheitis | 1 (1) | 1 (2) | 0 (0) | 0.42 |
| Delirium | 3 (3) | 1 (2) | 2 (4) | |
| ECMO | 1 (1) | 1 (2) | 0 (0) | |
| Mucus plugs | 2 (2) | 1 (2) | 1 (2) | |
| Pneumothorax | 1 (1) | 1 (2) | 0 (0) | |
| Pulmonary embolism | 1 (1) | 0 (0) | 1 (2) | |

Abbreviations: ECMO, extracorporeal membrane oxygenation; MV, mechanical ventilation; NIV, noninvasive ventilation; OPE, oscillation and pulmonary expansion. [a] Usual care included incentive spirometry and positive airway pressure therapy as needed. [b] OPE was delivered by the MetaNeb System (Hillrom). [c] $\chi^2$ test, *p* value < 0.05 considered significant.

## 4. Discussion

This retrospective study of health records evaluated OPE therapy as part of standard postoperative respiratory therapy for high-risk patients undergoing cardiac surgery. To our knowledge, this is the first attempt to study the effectiveness of OPE in this patient population. A previous study investigated this intervention for patients after general surgery [30].

The exact definition of postoperative pulmonary complications differs, just as reported rates of these complications vary from 2% to 40% [11]. One definition of postoperative pulmonary complications encompasses pulmonary infection, pleural effusion, bronchospasm and pneumothorax, chemical pneumonitis due to aspiration, atelectasis, acute respiratory distress syndrome, pulmonary edema, pulmonary embolism, and respiratory failure [33]. In our definition of respiratory complications, we also included the need for prolonged mechanical ventilation, need for noninvasive mechanical ventilation, and need for prolonged use of supplemental oxygen. This definition has been used in another study as well [30]. Before the OPE intervention, our postoperative pulmonary complication rate of 20% was comparable to rates described in other studies [6,34].

Although the underlying mechanisms responsible for postoperative pulmonary complications are most likely complex, atelectasis and decreased mucus clearance probably have an important role [35]. A low level of evidence exists that early postoperative mobilization, chest physiotherapy, and good oral hygiene may decrease postoperative pulmonary complications [35–39]. Similarly, a judicious and multipronged approach to analgesia, selective gastric decompression, and secretion mobilization may improve outcomes and are frequently used, but systemic evaluation of these interventions is lacking [35]. Among interventions shown to limit postoperative pulmonary complications, lung expansion therapies have some of the strongest evidence of beneficial effect [23]. Because OPE therapy can be started before extubation (as opposed to PAP with EzPAP), earlier intervention may help decrease the risk of prolonged ventilation and pulmonary complications.

Over the past several years, the need for improving patient outcomes and quality of care and using a value-based payment model have taken on increasing importance. Given this environment, it is especially important to decrease postoperative complications and improve quality of care. In fact, the need for postoperative mechanical ventilation for longer than 48 h and hospital LOS after major surgery represent quality measures that may be reportable to The Joint Commission and the Centers for Medicare & Medicaid Services [30].

In the current study, use of OPE was associated with a decreased rate of postoperative pulmonary complications from 20% to 10%, although the difference did not reach statistical

significance. Use of OPE was also associated with decreases in hospital and ICU LOS and with fewer cases of pneumonia and all respiratory tract infections. After multivariate adjustment for potential confounders, the ICU LOS was significantly shorter for patients after the OPE intervention.

We did not study the financial effect of this intervention. However, substantial savings can be achieved by decreasing ICU LOS and rates of postoperative pulmonary complications [40].

Our study exhibits several limitations. Firstly, the sample size was small, which may have impacted the robustness of our findings. Additionally, the absence of randomization in our study design poses a significant limitation. We did not employ a randomized controlled trial (RCT), a gold standard method for minimizing bias and establishing causal relationships between interventions and outcomes.

Furthermore, various interventions, such as a sedation 'vacation'/spontaneous breathing trial bundle and early mobilization, were implemented alongside our intervention. However, these were not systematically controlled for during the study, potentially confounding our results. Additionally, the retrospective nature of our investigation introduces the possibility of unidentified confounders influencing our findings.

Another limitation is our failure to adjust for seasonal variations, which could impact postoperative complications in cardiothoracic surgery [41]. Finally, the before-and-after design of our study inherently carries a risk of bias, which, despite efforts to mitigate, remains a concern [42].

## 5. Conclusions

Our study suggests a potential benefit of OPE therapy in reducing postoperative pulmonary complications among patients undergoing cardiac surgery. It is important to acknowledge the limitations inherent in our non-randomized study design. Despite efforts to compare two different practices, the absence of randomization introduces the possibility of bias and limits our ability to draw definitive conclusions regarding the effectiveness of OPE therapy.

Our findings suggest that among patients with higher ASA scores, who are at increased risk for postoperative complications, OPE therapy did not result in any identified adverse effects, particularly with regard to new or worsening pneumothorax.

To provide more robust evidence, future studies employing randomized controlled prospective models are warranted to confirm our findings. Additionally, further investigation into the use of OPE therapy across various postoperative patient populations is needed. This includes exploring its potential benefits for conditions such as chronic obstructive pulmonary disease and pulmonary contusions resulting from blunt chest trauma, both of which heighten the risk of postoperative pulmonary complications.

In conclusion, our study suggests a potential avenue for reducing postoperative pulmonary complications with OPE therapy. Further research is necessary to validate these findings and elucidate their precise role in improving patient outcomes. Specifically, the investigation into OPE therapy should extend beyond cardiac surgery patients to encompass a broader range of respiratory conditions that contribute to postoperative pulmonary complications.

**Author Contributions:** All authors made substantial contributions to the manuscript as follows: C.D.W.: study design, review of manuscript; K.M.H.: literature search, data collection; A.A.Z.: literature search, study design, review of manuscript; H.Z.A.-S.: literature search, study design, review of manuscript; A.S.: literature search, review of manuscript, manuscript preparation; A.R.R.: literature search, study design, review of manuscript; R.D.F.: contributed to design of analyses, analysis of data, provided interpretation of findings for primary author; A.S.Z.: literature search, study design, review of manuscript; M.A.R.: literature search, study design, data collection and analysis, manuscript preparation; PAP, positive airway pressure. All authors have read and agreed to the published version of the manuscript.

**Funding:** This research received no external funding.

**Institutional Review Board Statement:** IRB assessed and determined that this original study's activities did not necessitate review in accordance with the Code of Federal Regulations (45 CFR 46.102).

**Informed Consent Statement:** Not applicable.

**Data Availability Statement:** All relevant, deidentified data supporting the findings of this study are reported within the article.

**Acknowledgments:** The Scientific Publications staff, Mayo Clinic, provided editorial consultation, proofreading, administrative, and clerical support.

**Conflicts of Interest:** The authors declare that there are no conflicts of interest.

## Abbreviations

| | |
|---|---|
| ASA | American Society of Anesthesiologists |
| AVR | aortic valve replacement |
| CABG | coronary artery bypass graft |
| ICU | intensive care unit |
| IRB | Institutional Review Board |
| LOS | length of stay |
| MVR | mitral valve replacement |
| OPE | oscillation and pulmonary expansion |
| PAP | positive airway pressure |

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
