# Peer review of "Effect of Oscillation and Pulmonary Expansion Therapy on Pulmonary Outcomes after Cardiac Surgery†"

_2673-527X, doi:10.3390/jor4020008_

Round 1

Reviewer 1 Report

Comments and Suggestions for Authors

Abstract 

1- line 32: I didn't understand the information (July 1, 2019). I suggest withdrawing. 

2- line 33-35: The sentence needs to be rewritten. It's out of agreement. I believe you want to inform us about the findings that were defined for this study.

3- line 45: The acronym ICU needs to be defined.

Introduction

1- line 60: I suggest informing the ASA score physical status (minimum-maximum).

2- line 75-81: I suggest revising this sentence (perhaps presenting that although some studies point to a reduction in pulmonary complications ... This current study aims to .... and then, yes, finish with the hypothesis). 

Methods

1- line 128: The Figure needs to be assigned a number to ensure that it is correctly identified in the manuscript (Figure 1).

2- What is the guideline used to conduct this study? I didn't see any details on randomization. In case it didn't, I believe the TREND (Transparent Reporting of Non-randomized Designs) can be reported. Thus, it is worth stating in the methodology and results to provide a flowchart describing the configuration of the participants.

Results 

1- line 153-155: A flowchart describing the configuration of the participants should be presented (Figure 2).

2- In the legend of Tables 1, 2, 3 and 4, inform the significance value adopted (p value).

Author Response

Abstract 

  • line 32: I didn't understand the information (July 1, 2019). I suggest withdrawing. 

Response: Thank you for the feedback, portion suggested removed.

  • line 33-35: The sentence needs to be rewritten. It's out of agreement. I believe you want to inform us about the findings that were defined for this study.

Response: The sentence has been changed to “The primary outcome measure was the occurrence of severe postoperative respiratory complications, including the need for antibiotics, increased use of supplemental oxygen, and prolonged hospital length of stay (LOS). Demographic, clinical, and outcome data were compared between patients receiving usual care (involving post-extubation hyperinflation) and those treated under the new OPE protocol.

  • line 45: The acronym ICU needs to be defined.

Response: Acronym for ICU defined.

Introduction

  • line 60: I suggest informing the ASA score physical status (minimum-maximum).

Response: The sentence restructured to the following; Other risk factors for pulmonary complications include older age, higher American Society of Anesthesiologists (ASA) Physical Status Classification score (ranging from ASA I- A normal healthy patient A normal healthy patient to ASA-VI A declared brain-dead patient whose organs are being removed for donor purposes), congestive heart failure, chronic obstructive pulmonary disease, smoking history, and severe (class 3) obesity

line 75-81: I suggest revising this sentence (perhaps presenting that although some studies point to a reduction in pulmonary complications ... This current study aims to .... and then, yes, finish with the hypothesis). 

Response:  sentence restructured to the following; While a prospective study suggested that aggressive treatment with OPE may reduce postoperative pulmonary complications and resource utilization in high-risk patients undergoing general surgery, including a small subset undergoing thoracic surgery, its impact after cardiac surgery remains unexplored in the literature. To address this gap, the current study aims to investigate whether OPE can effectively decrease the incidence of postoperative respiratory complications in high-risk patients undergoing cardiac surgery compared to those receiving standard care.

Methods

  • line 128: The Figure needs to be assigned a number to ensure that it is correctly identified in the manuscript (Figure 1).

Response:  correction made. Figure 1 labeled and added.

  • What is the guideline used to conduct this study? I didn't see any details on randomization. In case it didn't, I believe the TREND (Transparent Reporting of Non-randomized Designs) can be reported. Thus, it is worth stating in the methodology and results to provide a flowchart describing the configuration of the participants.

Response:  I think this is a good suggestion. We have added the comment “Consistent with principles of TREND reporting guidelines for Quasi-Experimental Study Designs” in addition to adding a relevant reference (Reference 31). We have also added a flowchart  describing the configuration of the participants (Figure 2)

Results 

  • line 153-155: A flowchart describing the configuration of the participants should be presented (Figure 2).

Response: We have also added a flowchart describing the configuration of the participants (Figure 2).

2- In the legend of Tables 1, 2, 3 and 4, inform the significance value adopted (p value).

Response:  This was an oversight on our part. Thank you for pointing this out. We have added the statement “p value <0.05 considered significant” to the legend of each of the tables.

Reviewer 2 Report

Comments and Suggestions for Authors

Thank you for the opportunity to review this interesting and well-written article.

The authors present the benefits identified with OPE therapy for post-operative management of cardiac surgery patients. As this group of patients is at high risk for post-op respiratory complications, interventions and strategies that minimize them are of high importance and there still is limited knowledge on them. However, a randomized-controlled study on the field would be of much higher scientific value as current results are methodologically and statistically weak. 

Author Response

The authors present the benefits identified with OPE therapy for post-operative management of cardiac surgery patients. As this group of patients is at high risk for post-op respiratory complications, interventions and strategies that minimize them are of high importance and there still is limited knowledge on them. However, a randomized-controlled study on the field would be of much higher scientific value as current results are methodologically and statistically weak. 

Response:

Response:  Thank you for your insightful feedback on our paper regarding OPE therapy for post-operative management of cardiac surgery patients. We acknowledge the importance of robust scientific methodology in validating the efficacy of interventions in high-risk patient populations such as post-operative cardiac surgery patients.

While our study provides valuable insights into the potential benefits of OPE therapy in minimizing post-operative respiratory complications, we agree that a randomized controlled trial would further strengthen the evidence base and provide more definitive conclusions. We appreciate your suggestion and will consider it for future research endeavors in this area.

In the meantime, we hope that our study serves as a foundation for further investigation and discussion on strategies to improve the post-operative care of cardiac surgery patients. This point was also added as a limitation in the discussion section . Thank you once again for your thoughtful critique, which helps guide the direction of our research efforts.

Reviewer 3 Report

Comments and Suggestions for Authors

This is a most interesting study as the authors try to evaluate the effectiveness of anew  device that offers both oscillation and lung expansion therapy in post cardiac surgery patients. It is well known that a percentage of these patients will face an extended period of hospitalization due to pulmonary complications. Respiratory phystiotherapy has been proven as a significant preventive strategy to such complication.

Major comments:

1.This is a non randomized study, yet the authors tried to compare two different practices. Although they state some limitations, this should be highlighted and conclutions shouldn't clearly state that complication can be decreased.

2. The authors should state whether the included patients were consecutive or not. If not, it would be of interest to have the number of excluded patients and the cause of it.

3. The new protocol was added to the use of incentive spirometry, so we can't say for certain that any positive effect was the result of OPE.

4. The authors state that they didn't note any adverse events, yet they don't describe what adverse events did they monitor, in the outcome section.

Minor comments:

1. Authors need to give refernces for the information given on the first lines of the introduction regardic most common complications.

Author Response

This is a most interesting study as the authors try to evaluate the effectiveness of a new device that offers both oscillation and lung expansion therapy in post-cardiac surgery patients. It is well known that a percentage of these patients will face an extended period of hospitalization due to pulmonary complications. Respiratory physiotherapy has been proven as a significant preventive strategy for such complications.

Major comments:

1.This is a non-randomized study, yet the authors tried comparing two practices. Although they state some limitations, this should be highlighted, and conclusions shouldn't clearly state that complications can be decreased.

Response: Thank you for your insightful feedback on our paper regarding OPE therapy for post-operative management of cardiac surgery patients. We agree that a randomized controlled trial would further strengthen the evidence base and provide more definitive conclusions. We appreciate your suggestion and will consider it for future research endeavors in this area. The discussion section and conclusion sections have been updated to take into account reviewer suggestion and acknowledge limitations/conclusions drawn by our paper.

  1. The authors should state whether the included patients were consecutive or not. If not, it would be of interest to have the number of excluded patients and the cause of it.

Response:  Included patients were consecutive. This has now been clarified in the “Methods” section, subsection, “Treatment regimen” which now reads:

“From March 1 through June 30, 2019, consecutive patients undergoing these procedures received either incentive spirometry after extubation according….”

  1. The new protocol was added to the use of incentive spirometry, so we can't say for certain that any positive effect was the result of OPE.

Response: Incentive spirometry was provided to all patients throughout the study period. Incentive spirometry was provided during usual care period (March 2019 to June 2019) using nursing protocol or EzPAP. During OPE therapy period (July 2019 to October 2019) patients continued to receive incentive spirometry but no longer received PAP therapy during OPE treatment. This is clearly described under “Treatment Regimen” heading.

  1. The authors state that they didn't note any adverse events, yet they don't describe what adverse events did they monitor, in the outcome section.

 Response: Please note that under the “Outcome Measures” section we detail the adverse events that patients were screened for. We state: “Postoperative respiratory complications that patients were screened for included the need for prolonged mechanical ventilation (>24 hours after postsurgical hospital admission), prolonged need for noninvasive positive pressure ventilation (>24 hours after hospital admission), prolonged increased oxygen requirements (>40% fraction of inspired oxygen or 5 L/min >24 hours after admission), and readmission to the ICU. Screening also included a diagnosis of pneumonia based on criteria31 consisting of new pulmonary infiltrate, new-onset fever, purulent sputum, leukocytosis, and increased oxygen requirements.”

Minor comments:

  1. Authors need to give references to the information given in the first lines of the introduction regarding the most common complications.

Response: Reference for introduction added.

Reviewer 4 Report

Comments and Suggestions for Authors

The authors performed a retrospective analysis to document their newly adopted oscilation PEP regime after cardiothoracic surgery and its effect on patient outcomes. It is a retrospective study that comes with its own limitations , however retrospective analyses are required in order to provide a base for prospective studies in this area. The paper is well written and the methodology is sound. The results are understandable and thoroughly discussed. I think the papen can be published. 

Author Response

The authors performed a retrospective analysis to document their newly adopted oscilation PEP regime after cardiothoracic surgery and its effect on patient outcomes. It is a retrospective study that comes with its own limitations , however retrospective analyses are required in order to provide a base for prospective studies in this area. The paper is well written and the methodology is sound. The results are understandable and thoroughly discussed. I think the papen can be published. 

Response: Thank you for taking the time to review our manuscript. We appreciate your thoughtful evaluation and feedback.